# Function of Host Protein Staufen1 in Rabies Virus Replication

**DOI:** 10.3390/v13081426

**Published:** 2021-07-22

**Authors:** Gaowen Liu, Congjie Chen, Ruixian Xu, Ming Yang, Qinqin Han, Binghui Wang, Yuzhu Song, Xueshan Xia, Jinyang Zhang

**Affiliations:** Research Center of Molecular Medicine of Yunnan Province, Faculty of Life Science and Technology, Kunming University of Science and Technology, Kunming 650500, China; liugaowen1314@163.com (G.L.); kmustccj@163.com (C.C.); xrx115309259@163.com (R.X.); qjyangming@163.com (M.Y.); qqhan10@kust.edu.cn (Q.H.); WangBH@kust.edu.cn (B.W.)

**Keywords:** rabies virus, STAU1, SH-SY-5Y cells

## Abstract

Rabies virus is a highly neurophilic negative-strand RNA virus with high lethality and remains a huge public health problem in developing countries to date. The double-stranded RNA-binding protein Staufen1 (STAU1) has multiple functions in RNA virus replication, transcription, and translation. However, its function in RABV infection and its mechanism of action are not clear. In this study, we investigated the role of host factor STAU1 in RABV infection of SH-SY-5Y cells. Immunofluorescence, TCID_50_ titers, confocal microscopy, quantitative real-time PCR and Western blotting were carried out to determine the molecular function and subcellular distribution of STAU1 in these cell lines. Expression of STAU1 in SH-SY-5Y cells was down-regulated by RNA interference or up-regulated by transfection of eukaryotic expression vectors. The results showed that N proficiently colocalized with STAU1 in SH-SY-5Y at 36 h post-infection, and the expression level of STAU1 was also proportional to the time of infection. Down-regulation of STAU1 expression increased the number of Negri body-like structures, enhanced viral replication, and a caused 10-fold increase in viral titers. Meanwhile, N protein and G protein mRNA levels also accumulated gradually with increasing infection time, which implied that STAU1 inhibited rabies virus infection of SH-SY-5Y cells in vitro. In conclusion, our results provide important clues for the detailed replication mechanism of rabies virus and the discovery of therapeutic targets.

## 1. Introduction

Rabies is an important zoonotic infectious disease caused by rabies virus (RABV), with a mortality rate of almost 100% [1]. Neurotropic RABV causes fatal encephalomyelitis after infection of the central nervous system, and approximately 60,000 people die from rabies worldwide each year, with more than 15 million people receiving post-exposure prophylaxis [2,3]. Although rabies is a preventable disease, its impact is increasing and to date remains a huge health concern in both developing and developed countries.

RABV is a non-segmented negative-strand RNA virus belonging to the genus Lyssavirus of the Rhabdoviridae family with a genome size of approximately 12 kb [4]. The viral genes encode five viral proteins, which are organized as follows: 3′-nucleoprotein (N, 450aa)-phosphoprotein (P, 297aa)-matrix protein (M, 202aa)-glycoprotein (G, 505aa)-large protein polymerase protein (L, 2128aa)-5′ [5,6]. The viral N, P and L proteins form a helical nuclear capsid with genomic RNA, which is surrounded by a membrane composed of two viral proteins, M and G, and host lipids, forming a complete viral particle similar to a bullet. The N protein is the most transcriptionally abundant protein during infection and is usually recognized in infected tissues in the form of inclusion bodies [7]. The N protein participates in the formation of the viral nuclear capsid and binds genomic RNA to form the ribonucleoprotein complex (RNP). This allows the cellular nuclease P protein to bind to viral genomic RNA and L protein in RNP by bridging the N protein [8]. M protein is a multifunctional protein (21–25 kDa) that interacts with ribonucleoprotein and G protein and contributes to the assembly and outgrowth of RABV. M protein is also involved in the regulation of viral replication and translation processes [9]. The G protein is the only surface protein of the viral particle that allows RABV to enter the nervous system from peripheral sites through interaction with host cell receptors and is a major contributor to viral pathogenicity [10]. The L protein is the largest protein in the virus and has multiple enzymatic activities in the synthesis and processing of viral RNA, performing the function of an RNA-dependent RNA polymerase [11].

Staufen is a double-stranded (dsRNA) microtubule protein binding protein. Staufen has two homologous proteins, Staufen1 (STAU1) and Staufen2 (STAU2), both involved in the Staufen-mediated mRNA decay pathway (SMD) [12]. Staufen genes are widely expressed in mammals; STAU1 has no tissue bias compared to STAU2, while STAU2 is most abundantly expressed in neural tissues, especially in the brain. In addition, the STAU1 gene encodes at least two isoforms—STAU1^63^ and STAU1^55^—and contains four dsRNA binding domains, of which dsRBD3 and dsRBD4 are responsible for binding dsRNA [13]. It is generally accepted that the STAU1-mediated mRNA degradation pathway (SMD) is a post-transcriptional regulatory mechanism in which STAU1 triggers rapid mRNA degradation by binding to the STAU1 binding site in the untranslated region (3′UTR) of cellular mRNAs with a highly ordered secondary or tertiary structure. New insights are now available about SMD. STAU1 also recognizes another type of SBS, a dsRNA structure formed by partial base pairing of complementary Alu elements within a 3′UTR sequence molecule or between a 3′UTR sequence and a long-stranded non-coding RNA (lncRNA) molecule [14]. In addition, the nonsense-mediated mRNA decay (NMD) factor UPF1 is also involved in the SMD pathway [14,15,16]. Studies demonstrate that STAU1 plays an important role in the translocation, translational regulation and decay of cellular mRNA molecules. Several studies have shown that STAU1 binds both the 5′ untranslated region (5′UTR) and the 3′ untranslated region (3′UTR) of cellular mRNA, but the former increases the efficiency of translational activity and increases the number of mRNA molecules containing multimers, while the latter triggers the rapid degradation of mRNA [17].

STAU1 has been reported to play an important role in the infection cycle of multiple RNA viruses, including HIV-1, infectious bursal disease virus (IBDV), influenza A virus, enterovirus 71 (EV-A71), Ebola virus, and hepatitis C virus (HCV) [18,19,20,21,22]. Several studies have shown that STAU1 plays an important role in the early stages of HCV infection. During this time, STAU1 binds to the 3′UTR of the HCV RNA genome and to the negative-stranded HCV RNA intermediate, promoting viral translation, replication and trafficking, but not participating in nucleocapsid assembly to form intact viral p replication articles [18,23]. STAU1 was also revealed to bind to HIV-1 genomic RNA and Gag precursor protein through its NC structural domain to facilitate the polymerization process of Gag protein, as well as to achieve the HIV-1 genomic RNA capsidization and viral particle assembly process [24,25,26]. STAU1 RBD2–3 specifically binds to the 5′UTR of EV-A71, which enhances IRES activity while recruiting more viral RNA from the complex of ribosomes and prolonging the stability of viral RNA, thereby facilitating the viral translation process [20]. STAU1 is also a cellular component of the EBOV envelope, and endogenous STAU1 binds to several components of the EBOV RNP [21]. It actively participates in the process of EBOV RNA synthesis and enhances the synthesis of EBOV RNA. STAU1 binds to the 3′ and 5′ untranslated regions of EBOV genomic RNA and is an important EBOV provirus host factor [21]. In a study of infectious bursal disease virus (IBDV) in chickens, it was shown that chicken STAU1 interacted with viral dsRNA to attenuate the function of chicken MDA5 and inhibited viral dsRNA-induced production of chicken IFN-β. In addition, chicken STAU1 interacted with VP3 to inhibit the transcriptional activity of the chicken IFN-β promoter, thereby promoting IBDV replication [22]. STAU1 has been reported to interact with the influenza A virus ribonucleoprotein (N) protein part of the RNP complex and is thought to contribute to viral RNA coating to nascent viral particles. Furthermore, STAU1 binds to NS1, leading to the dissociation of UPF1 from STAU1, which inhibits the STAU1-mediated mRNA decay (SMD) process and facilitates viral replication [19,27]. As mentioned above, host protein STAU1 is required for RNA virus infection and may interact with multiple components within the viral genome, participating in processes such as genome replication, viral protein translation, and even virion assembly [18,19,20,21,22], but it remains unclear whether STAU1 plays a role in RABV infection.

In the current study, we demonstrated that STAU1 is co-localized with viral N protein and is recruited to the Negri bodies (NBs), an important site of rabies virus transcription and replication, during RABV infection and is involved in the regulation of RABV replication. This result also provides an important theoretical basis for understanding the key factors of RABV viral pathogenicity and finding new RABV therapeutic targets.

## 2. Materials and Methods

### 2.1. Cell Culture and Virus Inoculation

Human neuroblastoma cells (SH-SY-5Y) and mouse neuroblastoma cells (N2a) were cultured in DMEM medium (supplemented with 10% fetal bovine serum (FBS), 1% penicillin, and 1% streptomycin) and maintained at 37 °C with 5% CO_2_. Rabies virus HEP-Flury strain were kept in our laboratory. When the N2a cell or SH-SY-5Y cell density reached about 60%, the medium was replaced and the cells were infected with HEP-Flury at a multiplicity of infection (MOI) of 0.01 TCID_50_/cell. For RABV infection experiments, replace the 2% DMEM fresh culture medium after 2 h of infection to avoid the adverse effect of the primary virus on the experimental results.

### 2.2. Construction of STAU1 Eukaryotic Expression Vector and Overexpression

The STAU1 gene was cloned into pEGFP-N3 and pCI neo vector. A total of 293 T cell total RNA was extracted by TRIzol and reverse transcribed to synthesize human first strand cDNA. Two pairs of specific primers for the amplification of STAU1 gene were synthesized by Tsingke Biotechnology Co., Ltd. (Kunming, China), and the underlined enzyme cut sites (XhoI, BamHI and XhoI, XbaI) with the following sequences: F: 5′-CCGCTCGAGGGATGAAACTTGGAAAAAAACC-3′; R: 5′-CGCGGATCCGCACCT CCCACACACAG-3′; F: 5′-CCGCTCGAGCGGATGAAACTTGGAAAAAAACC-3′; R: 5′-GCTCTAGAGCTCAGCACCTCCCACACACAG-3′. The PCR-amplified STAU1 gene (containing BamHI, XhoI) was ligated to the pEGFP-N3 vector, and the STAU1 gene (containing XhoI, XbaI) was ligated to the pCI neo vector. The pEGFP-STAU1 and pCI-STAU1 plasmid DNA was extracted using the endotoxin-free plasmid kit, and it was transfected into SH-SY-5Y cells using transfection reagent (Bio BEST, Tianchang, China). After transfection for 48 h, the effect of STAU1 overexpression on RABV replication was verified by TCID_50_, Western blot assays.

### 2.3. Establishment of STAU1 Gene Interference in SH-SY-5Y Cell Line

STAU1-shRNA lentiviral particles (sc-76586-V) and control shRNA lentiviral particles were purchased from Santa Cruz (CA, USA) and infected SH-SY-5Y cells. Cells were maintained in 2% DMEM medium at 37 °C with 5% CO_2_ for 48 h, after which they were replaced with fresh medium containing 1 μg/mL puromycin. Cells were trypsin digested, passaged, and supplemented with puromycin-containing medium every 2–3 days. After all the cells in the negative control without lentivirus were dead and the cells infected with lentivirus were grown, a portion of the cells were washed once with PBS, and the supernatant was discarded as much as possible. A measure of 200 μL of RIPA lysis solution (strong) was added, the cells were lysed on ice for 30 min, and then the supernatant was collected by centrifugation at 12,000 × *g* rpm for 5 min for Western blot. The cell lines were tested for STAU1 expression. After the cell lines were successfully constructed, the cells were frozen in liquid nitrogen for seed preservation. Afterwards, STAU1 knockdown cell lines were infected with RABV and the effect of STAU1 downregulation on RABV replication was verified using Western blot, TCID_50_, and RT-qPCR.

### 2.4. Immunofluorescence Assay

Cell culture supernatants were discarded from 24-well plates or BeyoGlod^TM^ 35 mm confocal culture dishes, washed three times with phosphate-buffered saline (PBS), and SH-SY-5Y cells were fixed with methanol/acetone (1:1) and maintained at −20 °C for 20 min. In brief, the SH-SY-5Y cells were blocked with 500 μL of 5% skim milk and then incubated for 2 h at 37 °C with anti-RABV-N protein monoclonal antibody (1N1), anti-STAU1 polyclonal antibody (1:200). Then, the cell slides or BeyoGlod^TM^ 35 mm confocal dishes were washed three times with 500 μL of PBS and incubated for 1 h at 37 °C with FITC fluorescence-labeled goat anti-mouse secondary antibody and Alexa 594-fluorescence-labeled goat anti-rabbit secondary antibody (1:200) diluted in 5% skim milk. SH-SY-5Y cell nuclei were stained using DAPI (1:1000) diluted in blocking buffer and incubated at 37 °C for 15 min. After 15 min, the SH-SY-5Y cell were washed with PBS and observed under confocal microscope. Afterwards, the rabies-induced structures were quantified using the cell counting function of Image J software. In the Analyze Particles window, the parameters were set as: size (pixel^2^): 50-Infinity; circularity: 1.00, and other parameters with default values. In order to ensure the reliability of the results, we took two different measures: 1) we counted the cell concentrations before the cell plating, and the Staufen 1 gene knockdown cells and the control cells were put into the dish in the same number (1 × 10^4^); 2) we randomly selected five different visual fields for statistical analysis in the experimental group and the control group during observation. In addition, we used the same exposure time and contrast for the experimental group and the control group in the laser confocal observation.

### 2.5. TCID_50_ Assay

Briefly, N2a cells were uniformly inoculated into 96-well plates with 2% DMEM medium. RABV was diluted from 10^−1^–10^−8^ with 2% DMEM medium and 100 μL/well was incubated with cells for 48 h. After 48 h of infection, immunofluorescence assay was performed and the number of wells with cell fluorescence in each dilution gradient was counted. The Reed–Muench method was used to calculate a 50% tissue culture infective dose (TCID_50_). lgTCID_50_ = proportion distance (PD) × difference between the logarithm of dilution + logarithm of dilution above a 50% lesion rate. PD = (% infected cultures above 50%) − 50/(% infected cultures above 50%) − (% infected cultures below 50%).

### 2.6. Western Blot

The cells were lysed in RIPA lysis buffer to collect the protein, which was quantified using the Micro BCA Protein Assay Reagent. The protein lysates (20 μg) were separated by 12% SDS poly-acrylamide gel electrophoresis (SDS-PAGE) and transferred to nitrocellulose filter membrane NC, (Pall, Shanghai, China) with a pore size of 0. 22 μm. Membranes were blocked in PBST with 5% skimmed milk for 2 h in 37 °C. Following this, the primary antibodies (Anti-STAU1 (Abcam, Cambridge, MA, USA), Anti-RABV-N (1N1) (produced by lab)and Anti-GAPDH (Goodhere, Hangzhou, China)) were incubated at 37 °C for 2 h or 4 °C for overnight, washed thrice with PBST for 3–5 min, then incubate with secondary antibody coupled to horseradish peroxidase for 1 h at 37 °C. Membranes were washed again and analyzed by chemiluminescence using the ECL^+^ kit.

### 2.7. Total RNA Extraction, Reverse Transcription Quantitative Real-Time PCR

Total RNA was isolated from the cells using TRIzol (Invitrogen, Carlsbad, CA, USA) reagent. Random primers, reverse transcriptase (Vazyme Biotech Co., Ltd., Nanjing, China) and 3 µg of total RNA were used to synthesize cDNA. Quantification of STAU1 mRNA, RABV-N mRNA and RABV-G mRNA was performed by real-time quantitative PCR (qPCR) using STAU1 forward and reverse primers: 5′-GCTCACTCAGACACACATTGGG-3′ and 5′-GGTCACGCTGAGTAGGAA GC-3′; RABV-N forward and reverse primers: 5′- AGCAGCAATGCAGTTCTTTGAGGG-3′ and 5′- TTGTCAGTTCCATGCCTCCTGTCA-3′; RABV-G forward and reverse primers: 5′-AGCAGAGGGAAGAGAGAGCATCCAAA-3′, and 5′-ATCGCGACCCATGTTCCATCCATA-3′, respectively; GAPDH forward and reverse primers: 5′-GTCTCCTCTGACTTCAACAGCG-3′ and 5′-ACCACCCTGTTGCTGTAGCCAA-3′. QPCR analysis was repeated in an ABI 7500 qPCR system using SYBR green master mix (Vazyme Biotech Co., Ltd., Nanjing, China).

### 2.8. Statistical Analyssis

Relative quantitative analysis of Western blot results and fluorescence signal maps were obtained using Image J software. TCID_50_ results were obtained using SPSS software (Version.25), while ANOVA results were obtained with ± SD representation; *p* < 0.05 indicated a statistically significant difference.

## 3. Results 

### 3.1. STAU1 Is Recruited to RABV Negri Bodies (NBs)

The Negri bodies (NBs) are the processing plant of rabies virus—an important production site for transcription and replication, first discovered in 1903. During RABV infection of cells, a large number of host proteins are recruited into Negri bodies (NBs), such as CCTγ, CCTα, and Prefoldin 1, which have an impact on virus replication. STAU1, an important host protein, plays an important role in the replication of several viruses, and it has been shown that STAU1 is recruited into EBOV inclusion bodies. To determine whether STAU1 is recruited into RABV Negri bodies (NBs) upon viral infection, we infected SH-SY-5Y cells with the RABV HEP-Flury strain for 36 h and visualized the localization of endogenous STAU1 by immunofluorescence analysis (IFA).

In non-RABV-infected cells, STAU1 was uniformly distributed in the cytoplasm. When SH-SY5Y cells were infected with RABV, the RABV N protein was concentrated in the “factory” of RABV replication, the Negri bodies (NBs), at which time the host protein STAU1 also aggregated to and co-localized with the viral N-protein (Figure 1). This result indicates that when the host cell is infected by RABV, the host protein STAU1 co-localizes with the RABV N in the Negri bodies (NBs).

### 3.2. STAU1 Is Involved in Host Resistance to RABV Replication

Previously, we confirmed that STAU1 is co-localized in the Negri body. The next question to ask is whether STAU1 affects RABV replication. To achieve our aim, we constructed a STAU1 overexpression cell line. pEGFP-STAU1 was successfully transfected into SH-SY-5Y cells (40% transfection efficiency) and the STAU1 gene was successfully expressed (Figure 2A). To investigate the effect of STAU1 upregulation in host cells on RABV replication, culture supernatants and cells were collected 24 h after cells were transfected with the overexpression plasmid and 48 h after cells were infected with RABV. TCID_50_ was used to assess the titer of virus in cell supernatants, and Western blot was used to detect the expression of STAU1 and N protein in cells (Figure 2B–D). The results showed that there was no significant difference in STAU1 overexpression on RABV replication, and there was no significant difference in the total amount of RABV N protein and virus titer in the supernatant compared with control. In addition, in order to exclude the effect of GFP on STAU1 function, we constructed the pCI-neo-STAU1 eukaryotic expression vector. After STAU1 expression was upregulated; viral titers, STAU1 and RABV-N expression were detected, consistent with previous methods (Appendix A). The results were not different from the previous.

To further confirm the function of STAU1 in RABV replication, we constructed LV-shRNA-iSTAU1 cell line and performed RABV infection assay. The results of Western blot assay showed that SH-SY-5Y cells were significantly infected by RABV and the expression level of endogenous STAU1 protein was downregulated by 60%, while the expression level of viral N protein was upregulated by 100% and P protein was upregulated by about 30% (Figure 3A,B). The results of the TCID_50_ assay for viral titer in cell culture supernatants showed that the downregulation of host STAU1 levels enhanced viral replication, and the viral titer was approximately 10 times higher than that of the control group (Figure 3C). To further validate this phenomenon, the levels of host STAU1 transcripts and the gene encoding the RABV protein were examined using RT-qPCR in RABV-infected STAU1-interfering cell lines. RT-qPCR results showed that the mRNA levels of STAU1 were reduced by approximately 60% compared to the control group. RABV N protein gene increased by about 40% and RABV G protein gene increased by about 50% (Figure 3D).

In general, the RT-qPCR results were more consistent with the Western blot results. These results suggest that down-regulation of endogenous STAU1 protein and transcript levels in host cells increases viral replication in the host when infected by RABV, indicating a possible involvement of STAU1 in host resistance to RABV replication.

### 3.3. STAU1 Interferes with the Formation of Negri Bodies

In this study, we found that STAU1 was recruited to Negri bodies (NBs) in RABV-infected SH-SY-5Y cells and that RABV viral titers and mRNA transcript levels were increased after downregulated of STAU1. Negri bodies are the viral factories of RABV and affect viral replication. To reveal the reason why STAU1 downregulation leads to enhanced viral replication, we analyzed the involvement of STAU1 in the formation of NBs in STAU1 knockdown experiments. We examined the subcellular distribution of STAU1 and RABV-N in STAU1-knockdown cell line (LV-shRNA-STAU1) and control (LV-shRNA-iC) using IFA (Figure 4A,B). Similar to the previous results, STAU1 co-localized with RABV-N in Negri bodies. Interestingly, RABV N staining was stronger in cells with relative less STAU1 expression, showing stronger red fluorescence, implying higher RABV-N expression. Therefore, we randomly selected five sets of graphs for counting statistics of Negri body-like structures. We have distinguished inclusions (range size > 50 pixels) (Figure 4C, Appendix A), and the results showed that the number of Negri body-like structures was higher after STAU1 interference compared to the control.

In order to explore the mechanism of STAU1 blocking RABV replication, we collected cell samples after infection with RABV at different time points and performed RT-qPCR to analyze the transcript levels of N protein, G protein and STAU1. The results showed that the transcripts of viral N and G proteins showed an exponential increase during the infection period of 12–72 h (Figure 5A,B). At the same time, STAU1 transcripts were upregulated (Figure 5C).

## 4. Discussion

In the endless war between hosts and viruses, both sides of the struggle revolve around host factors. Negri bodies (NBs)—rabies virus replication factories—not only have viral capsids, phosphoprotein and L-proteins, but also contain a range of host factors such as endothelial nitric oxide synthase (eNOS), TLR3, Hsp70 and focal adhesion kinase (FAK) [28,29,30,31,32]. In recent years, there has been increasing interest in the composition of the Negri bodies (NBs) and especially the interaction with host factors during RABV replication. The role of some host factors in anti-rabies virus defense has been well demonstrated; e.g., the interferon-stimulated gene 15 (ISG15) with anti-RABV function and its activating enzyme UBA7, CCTα, CCTγ and Prefoldin are important host factors for RABV replication and transcription [33,34,35,36]. However, the interrelationships between these host factors and viral proteins, and the detailed mechanisms of their roles in the viral life cycle have not been systematically elucidated due to the failure to comprehensively elucidate the host factors interacting with RABV. Therefore, the search for new host factors is important for studying the replication mechanism of RABV and its treatment.

STAU1 is a highly conserved double-stranded RNA-binding protein found in the majority of bipartite animals, involved in translational regulation and cellular RNA translocation and degradation [37]. Its function has been demonstrated in RNA viruses, such as HIV-1, HCV and influenza. These results undoubtedly bring a boon to the study of STAU1 in rabies virus, which is also an RNA virus with nucleic acid localization, transcription and replication occurring centrally in the Negri bodies (NBs) structure. In our study, we found that RABV infection of SH-SY-5Y cells resulted in the recruitment of host factors to the Negri bodies (NBs) and co-localization with the RABV N protein, which is an essential component of the viral ribonucleoprotein (RNP) complex, formed by the N protein encapsulating the viral genomic RNA. This result is similar to that of Ebola virus, where EBOV VP35 interacts with STAU1 to co-localize in the EBOV envelope [21]. This implies that STAU1 may be involved in the formation of RABV RNP.

To further verify the function of STAU1 in RABV-infected SH-SY-5Y cells, we overexpressed the STAU1 gene and interfered with the STAU1 gene, respectively. Although the result of the effect of overexpression of STAU1 on RABV replication was not what we expected, it is understandable that it caused this phenomenon. As STAU1 mainly functions through the SMD pathway, which requires multiple proteins to interact with STAU1 and function together, so far, several cytokines including STAU1, Upf1 and PNRC2 are considered to be required for efficient SMD [13,38]. Therefore, single overexpression of STAU1 gene did not show significant results.

Compared to STAU1 overexpression experiments, we performed functional validation of STAU1 in RABV replication by interfering with STAU1 expression, and the results were more convincing. The results showed that down-regulation of STAU1 expression can up-regulate the replication of RABV. In addition, there was a significant increase in Negri body-like structures. Although the fluorescence intensity of the controls was also high, the fluorescence showed a scattered distribution and they were mainly RABV RNPs. RNPs in NBs are expelled through a cytoskeleton-independent mechanism and further transported along microtubules to form new viral particles or secondary viral factories [39,40]. This result implies that STAU1 affects the formation of RABV Negri bodies, with the opposite trend of Toll-Like Receptor 3 (TLR3) [30]. In addition, studies have shown that NBs have characteristics of liquid organelles; they are spherical and fuse to form larger structures. These organelles formed during viral infection concentrate viral proteins, cytokines and nucleic acids, establishing a platform to promote viral replication [39]. Thus, the formation of Negri body-like structures is favorable for RABV replication. Meanwhile, we found that STAU1 mRNA levels increased during infection. We speculate that the up-regulation of STAU1 transcript levels may be a result of the host’s antiviral response or stress response. Although a similar regulatory pathway was not reported in STAU1, its homolog STAU2 could act as a positive regulator that interacts with HIV-1 Rev to promote HIV-1 infection [41]. These results suggest that STAU1 was indeed involved in the process of inhibiting the formation of the RABV RNP complex and further inhibiting viral replication.

Although our experimental results confirm that STAU1 plays a non-negligible role in RABV replication, the SMD pathway mediated by STAU1 during RABV replication is still unclear. As a result of numerous studies, STAU1 mainly recognizes two types of STAU1 binding sites (SBS): one is a highly ordered secondary or tertiary structure in the 3′ untranslated region (3′UTR) of mRNA, and the other is formed by base pairing between two RNA molecules, one Alu sequence in the 3′UTR of the target mRNA and the other Alu sequence in the long-stranded non-coding RNA (lncRNA) [14]. We are aware that several positive-stranded RNA viruses, such as HIV-1, dengue virus (DENV) and West Nile virus (WNV), can encode microRNAs similar to long non-coding RNA (lncRNA) to regulate viral replication [42,43]. Rabies virus, a negative-stranded RNA virus, does not produce viral miRNAs, and instead produces small ncRNAs called viral leader RNA (leRNAs) [44]. Studies have confirmed that leRNAs interact with the host Hsc70 (HSPA8) to inhibit rabies virus replication [44]. In addition, UPF1, an important component of SMD, may also be involved in SMD regulation [12,13]. To explore the potential connection between STAU1, HSPA8 and UPF1, we performed an interaction analysis using the STRING database, and the results showed a possible interaction between HSPA8 and UPF1 (Figure 6A). The interaction between STAU1 and UPF1 has been demonstrated in influenza viruses that are also negative-strand RNA viruses, and it has been shown that influenza virus NS1 inhibits STAU1-mediated mRNA degradation by interfering with the binding of UPF1 to STAU1 [45].

In summary, we predicted a hypothetical model for STAU1-mediated RABV mRNA degradation (Figure 6B). When RABV transcribes a large amount of mRNA, the host cell immune response produces a large amount of lncRNA [46] and binds to the 3′ end of RABV mRNA, which also has a large amount of Alu at the 3′ end, and a large amount of host STAU1 recognizes the RABV mRNA as a non-coding sequence, which binds to the formed double-stranded structure and recruits UPF1 protein, which in turn recruits HSPA8 protein to the 3′ end of lncRNA to stabilize the structure, thereby regulating RABV mRNA degradation and inhibiting RABV replication. In addition, when HSPA8 protein is recruited, less HSPA8 protein binds to leRNA, more leRNA is present and interferes with the binding of RABV nuclear protein to genomic RNA, further inhibiting RABV replication. Conversely, when STAU1 protein was reduced, RABV replication was promoted.

Although we found that STAU1 inhibited rabies virus replication and proposed a plausible mechanism for SMD degradation of RABV mRNA, the dissociation of STAU1 from UPF1 mediated by the interaction of viral proteins with STAU1 needs to be supported by robust experimental data.

## 5. Conclusions

This study reveals that the host protein STAU1 was recruited to the NBs in RABV-infected cells and may play a role in host antagonism against rabies virus proliferation.

## Figures and Tables

**Figure 1 viruses-13-01426-f001:**
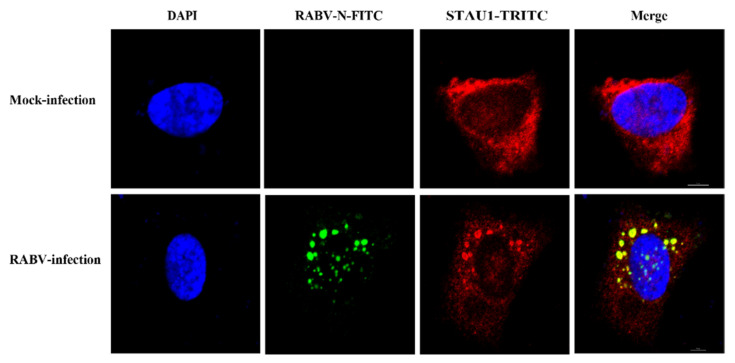
Subcellular distribution of STAU1 in RABV infected SH-SY5Y cells. Note: The scale is 5 μm.

**Figure 2 viruses-13-01426-f002:**
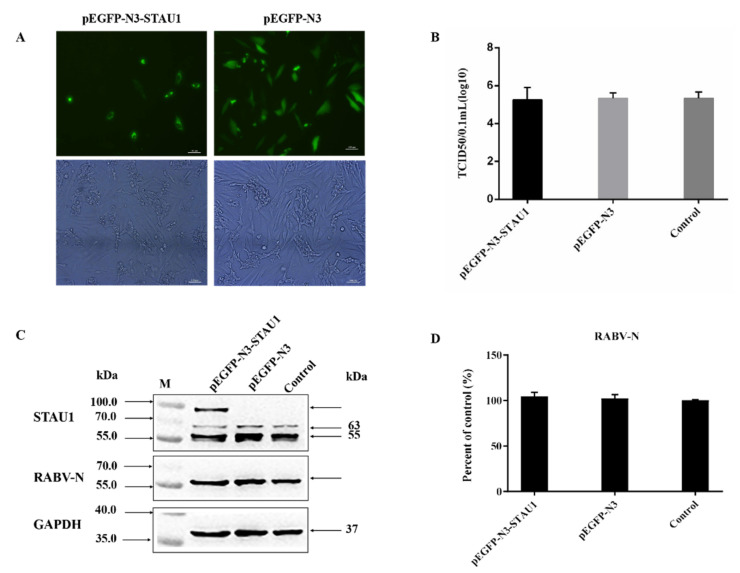
The effect of STAU1 upregulation on RABV replication. The pEGFP-STAU1 plasmid DNA was transfected with SH-SY-5Y cells. (**A**) The overexpression of STAU1 in SH-SY-5Y cells was observed by fluorescence microscopy, at 48 h post-transfection. Bar = 100 μm. (**B**) TCID_50_ determination of RABV after upregulating STAU1. (**C**) Overexpression of STAU1 detected by Western blot and its effect on RABV N protein expression at 48 h post-transfection. (**D**) RABV-N protein expression was normalized by comparison with control as 100% in Figure 2C.

**Figure 3 viruses-13-01426-f003:**
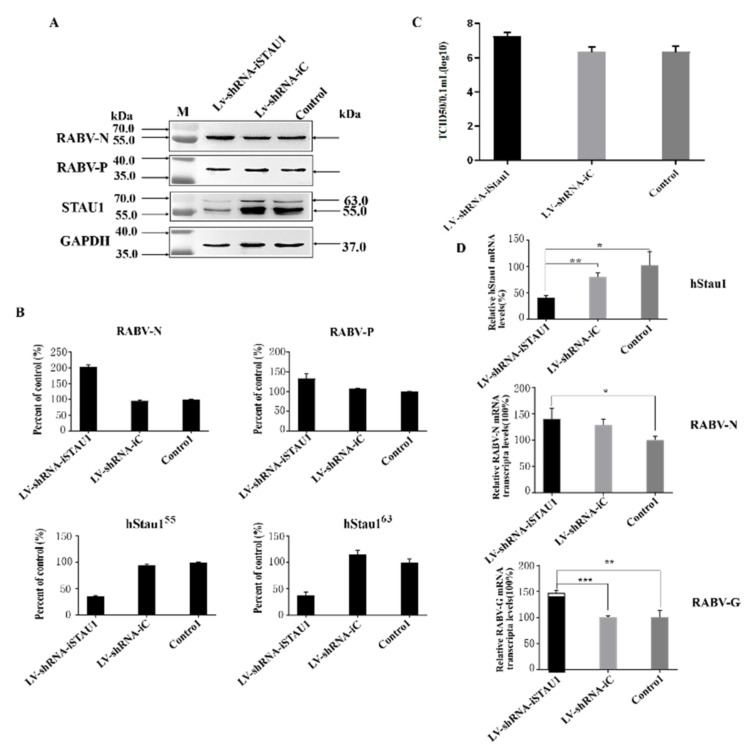
The effect of STAU1 downregulation on RABV replication. (**A**) Western blot to detect STAU1 expression in STAU1 knockdown cell lines and the effect of STAU1 downregulation on RABV-N, RABV-P expression, 48 h post-infection with RABV. (**B**) RABV-N, RABV-P and STAU1 expression was normalized by comparison with control as 100% in Figure 3A. (**C**) Detection of TCID_50_ of RABV after cells down-regulated STAU1. (**D**) RT-qPCR detection of STAU1 genes, virus N and G transcript changes after down-regulated STAU1 and infection of RABV. *, *p* < 0.05; **, *p* < 0.01; ***, *p* < 0.001. Note: In the figure, two homologous proteins of STAU1 are represented by hStau1^55^ and hStau1^63^ respectively; hstau1 is short for human Staufen 1 protein.

**Figure 4 viruses-13-01426-f004:**
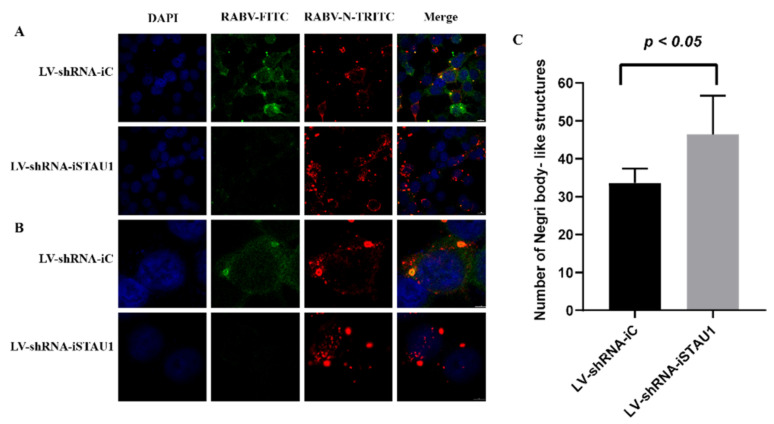
STAU1 affects the formation of Negri body-like structures. (**A**) Subcellular distribution of STAU1 in RABV-infected LV-shRNA-iSTAU1/LV-shRNA-iC cell line, at 48 h post-infection, Bar = 10 μm. (**B**) Enlarged area of A panels (white box), Bar = 5 μm. (**C**) Number of Negri body-like structures. Five sets of fluorescence images were randomly selected (Figure 4A, red), and image J counted the number of Negri body-like structures (range size > 50 pixels).

**Figure 5 viruses-13-01426-f005:**
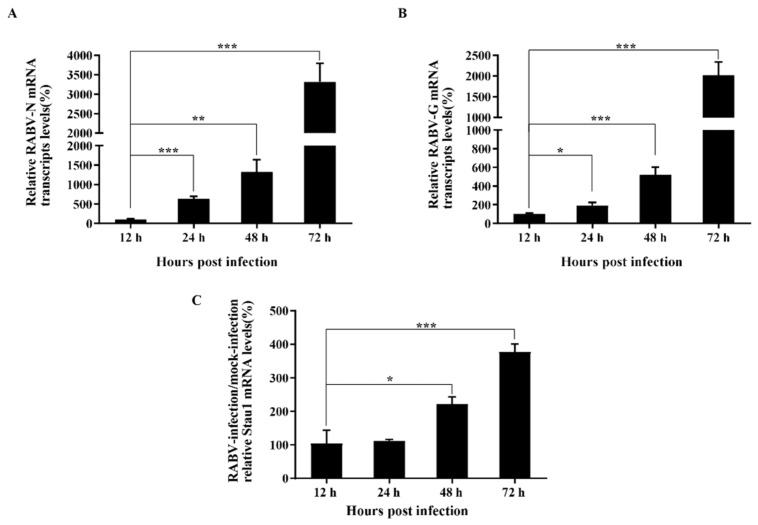
Analysis of dynamic changes in viral protein-encoding genes and STAU1 transcripts. (**A**) RABV-N mRNA transcripts levels. (**B**) RABV-G mRNA transcripts levels. (**C**) STAU1 mRNA transcripts levels at different times post-infection. Note: *, *p* < 0.05; **, *p* < 0.01; ***, *p* < 0.001.

**Figure 6 viruses-13-01426-f006:**
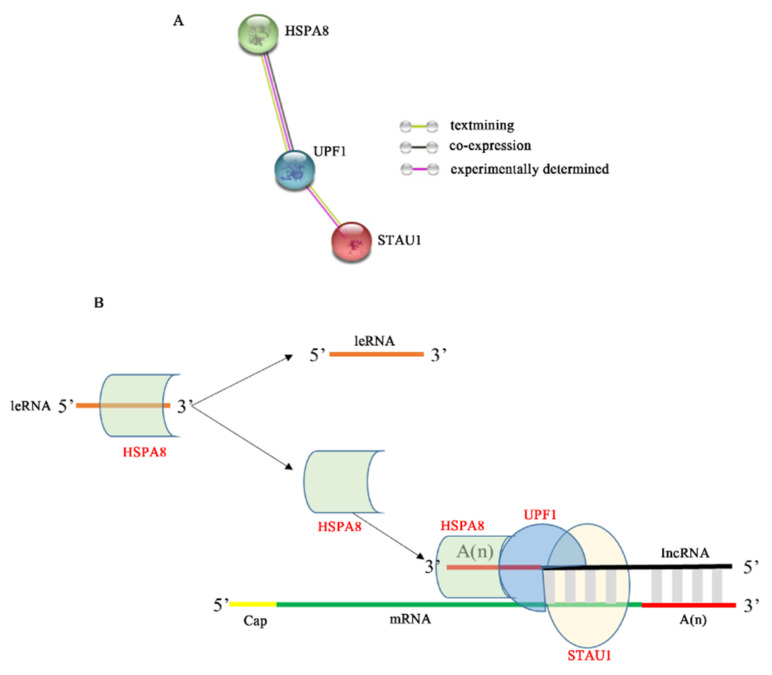
Mechanism model of STAU1 inhibiting RABV replication. (**A**) HSPA8, UPF1, and STAU1 protein interactions. (**B**) The model of STAU1 inhibits RABV replication.

## Data Availability

Not applicable.

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
