# Peer review of "Function of Host Protein Staufen1 in Rabies Virus Replication"

_viruses, 2021, doi:10.3390/v13081426_

Round 1

Reviewer 1 Report

The authors have added some new data to help support their assertion that Staufen 1 affects RABV replication. The new data increases the papers soundness and impact.

There are a number of changes that the authors should make to the text before it is accepted for publication.

Results:

Line 221 (Figure 1): Because Staufen1 colocalizes with negri bodies does not mean it plays a role in RABV replication. This part of the sentence should be removed as it really doesn't help the paper in any way. The later data supports this and such assertions should be saved for that data, not here.

Figure 2A: Please make sure the bright field matches the fluorescence panel, the transfection efficiency should be reported in the text as this could severely impact the results of this assay.

Lines 280-281: "Interestingly, RABV was more sensitive to cells with relative low STAU1 expression..." This sentence doesn't make much sense...maybe instead replace with 'Interestingly, RABV N staining was stronger in cells with relative less STAU1 expression.'

Throughout the text the authors refer to 'Negri bodies-like structures', this should be corrected to 'Negri body-like structures'.

Lines 283-285: More details on how cells were selected should be provided, here or in the Methods section, as in the Supp Figures it appears  there was infection in cells with little or no staining for STAU1 in both the WT and KN cells. Were whole views counted or were cells with no STAU1 present in the KD control excluded so as not to affect the relative quantification between the WT and KD conditions?

Figure 4: please include a thin white box-line in one of panels in A to identify which region has been magnified in B and include reference to this in the legend to guide the readers better.

Lines 297-298: This assertion should be removed as there is no data supporting this. The possibility is now raised as conjecture in the Discussion and so should be remove from the results section.

Discussion:

Line 310: "the on of" should be corrected

Lines 338-342: This section should be moved to Results to support the gfp-staufen1 data. Construction of this vector should also be described in the in the Methods section.

Line 353: ",with a similar and opposite trend of TLR3"...this does not make sense, how can something be similar and opposite, please clarify what is meant by this statement.

Line 360: ",or may be induced by the virus to promote its own proliferation." But the point of the paper is that STAU1 inhibits replication have I missed something? Consider removing or explaining better what is meant here.

Line 379: String analysis does not really prove anything, it just looks for potential relationships. This should be rephrased.

Line 386: This is a hypothetical model, the authors should rephrase, editors should be able to help with this.

Methods:

1) The authors interchange between knockout and knockdown in the description of their shRNA expressing cell lines throughout the paper, not just in the methods. These are Knock down cells, and reference to 'knockout' should be removed/corrected.

2) The authors should include a description of how they made the new Staufen1 expressing plasmid in the the plasmids section. 

3) The authors should include a section on how the analysis of negribodies in wt and knockdown cells was performed. This could be included in the IFA section of the methods.

Author Response

Dear Editors ,

Thank you for your comments concerning our manuscript entitled “Function of host protein Staufen1 in rabies virus replication “(viruses-1270057). Those comments are all valuable and very helpful for revising and improving our paper, as well as the important guiding significance to our research. We have studied comments carefully and have made the correction which we hope meet with approval. The revised portion is marked in red in the paper. The main correction in the paper and the responses to the reviewer’s comments are as following:

Result:

Line 221 (Figure 1): Because Staufen1 colocalizes with negri bodies does not mean it plays a role in RABV replication. This part of the sentence should be removed as it really doesn't help the paper in any way. The later data supports this and such assertions should be saved for that data, not here.

Answer: Very thank the reviewer’s comments. We apologize for the deviation in our understanding. We have deleted“plays a role in the replication of RABV” this part of inappropriate expression in the manuscript.

Figure 2A: Please make sure the bright field matches the fluorescence panel, the transfection efficiency should be reported in the text as this could severely impact the results of this assay.

Answer: We are glad to hear your suggestion. In the revised manuscript, we carefully checked the original data, confirmed that the bright field matched the fluorescence panel, and described the transfection efficiency.

Lines 280-281: "Interestingly, RABV was more sensitive to cells with relative low STAU1 expression..." This sentence doesn't make much sense...maybe instead replace with 'Interestingly, RABV N staining was stronger in cells with relative less STAU1 expression.'

Answer: Thanks for your suggestion. In the revised manuscript, we replaced the original sentence with “Interestingly, RABV N staining was stronger in cells with relative less STAU1 expression”.

Throughout the text the authors refer to 'Negri bodies-like structures', this should be corrected to 'Negri body-like structures'..

Answer: We are very sorry for our incorrect writing. It has been corrected in the revised manuscript.

Lines 283-285: More details on how cells were selected should be provided, here or in the Methods section, as in the Supp Figures it appears there was infection in cells with little or no staining for STAU1 in both the WT and KN cells. Were whole views counted or were cells with no STAU1 present in the KD control excluded so as not to affect the relative quantification between the WT and KD conditions?

Answer: We really appreciate your suggestions, which are very helpful to the perfection of our manuscript. In the method section, we describe the analysis method of Negri bodies in WT and Knockdown cells in detail. Stau1 staining results in WT and KN cells were considered to be normal. In knockdown cell lines, the fluorescence brightness was significantly lower than that in the control group at the same exposure time and contrast. In order to ensure the reliability of the results, we took two different measures. On the one hand, we counted the cells concentrations before the cell plating, and the Staufen 1 gene knockdown cells and the control cells were put into the dish in the same number (1x10^4); On the other hand, we randomly selected 5 different whole views for statistical analysis in the experimental group and the control group during observation. In addition, we used the same exposure time and contrast for the experimental group and the control group in the laser confocal observation.

Figure 4: please include a thin white box-line in one of panels in A to identify which region has been magnified in B and include reference to this in the legend to guide the readers better.

Answer: We are glad to hear your suggestion. We have reannotated the picture as you suggested in the revised manuscript.

Lines 297-298: This assertion should be removed as there is no data supporting this. The possibility is now raised as conjecture in the Discussion and so should be remove from the results section.

Answer: We really appreciate your suggestions.which again provides favorable evidence for the inference that STAU1 is involved in resisting viral replication” was deleted.

Discussion:

Line 310: "the on of" should be corrected

Answer: We are very sorry for our incorrect writing. “the interaction with” replaced “the on of”.

Lines 338-342: This section should be moved to Results to support the gfp-staufen1 data. Construction of this vector should also be described in the in the Methods section.

Answer: We are glad to hear your suggestion. This section have been moved to Results to support the gfp-staufen1 data. And the description of pCI-STAU1 vector construction was added in 2.2.

Line 353: ",with a similar and opposite trend of TLR3"...this does not make sense, how can something be similar and opposite, please clarify what is meant by this statement.

Answer: We are very sorry for our incorrect writing. “a similar and” was deleted.

Line 360: ",or may be induced by the virus to promote its own proliferation." But the point of the paper is that STAU1 inhibits replication have I missed something? Consider removing or explaining better what is meant here.

Answer: We are glad to hear your suggestion. We have deleted the “, or may be induced by the virus to promote its own proliferation” in the revised manuscript.

Line 379: String analysis does not really prove anything, it just looks for potential relationships. This should be rephrased.

Answer: We apologize for the inappropriate characterization. The original manuscript was replaced with "In order to explore the potential relationship between STAU1, HSPA8, and UPF1, we used the String database for interaction analysis". 

Line 386: This is a hypothetical model, the authors should rephrase, editors should be able to help with this.

Answer: Thanks for this valuable comment. We rephrase this part in revised manuscript.

 Methods:

The authors interchange between knockout and knockdown in the description of their shRNA expressing cell lines throughout the paper, not just in the methods. These are Knock down cells, and reference to 'knockout' should be removed/corrected.

Answer: We are very sorry for our incorrect writing. “knockout” was replaced with “knock down”.

The authors should include a description of how they made the new Staufen1 expressing plasmid in the the plasmids section.

Answer: We added a description of the PCI-STAU1 vector construction to the method.

The authors should include a section on how the analysis of negribodies in wt and knockdown cells was performed. This could be included in the IFA section of the methods.

Answer: Thanks for your suggestion, we have added the corresponding analysis method in the IFA section.  

Reviewer 2 Report

Line 47: genomic RNA is repeated

Line 309/310: should read '...composition of the Negri body and especially the on of host factors there during RABV... Answer: “the function of” replace with “the on of”,and ”in” was deleted

Otherwise it looks good.

Author Response

Thank you very much for your positive comments on our manuscript. Considering your comments, we have made the following changes.

Line 47: genomic RNA is repeated.

Answer: We are very sorry for our incorrect writing. Repeated “genomic RNA” was deleted.

Line 309/310: should read '...composition of the Negri body and especially the on of host factors there during RABV... Answer: “the function of” replace with “the on of”,and ”in” was deleted.

Answer: We are very sorry for our incorrect writing. In the revised manuscript,“the interaction with” replaced “the on of”.

This manuscript is a resubmission of an earlier submission. The following is a list of the peer review reports and author responses from that submission.

Round 1

Reviewer 1 Report

Liu et al present an analysis of a role for the RNA binding protein Staufen 1 in rabies virus replication. The authors conclude that Staufen 1 plays a role as an antiviral protein. Unfortunately, the data does not support the conclusions drawn.

Figure 1, the authors show localisation of Staufen 1 in Negri bodies during infection and conclude that Staufen 1 interacts with N protein and plays a role in Rabies virus replication (p5, line 205). This data does not support this conclusion.

Figure 2, the authors overexpress a GFP tagged version of Staufen 1 and show no effect on virus replication in any system tested. In the Discussion the suggest that the tag may affect Staufen 1 function. No data is presented to test this.

Figure 3, knock down of Stafuen 1 is performed and a small effect on viral replication is shown. As indicated by the authors Staufen 1 plays many roles in RNA metabolism. There is no evidence presented that the effect on virus replication is specific.

Figure 4, the authors show that Staufen 1 mRNA increases during infection and infer that Staufen 1 is being unregulated because it is an antiviral gene.  No evidence is presented to support this.

While the publication of "negative" data should be encouraged the presentation and interpretation of negative data should also be supported by the experiments contained in the paper, which in this case I don't believe they are.

Minor

Introduction reads like it was written in parts and assembled, Rabies virus and Negri bodies are abbreviated numerous times in succession. The second paragraph in the Introduction is not clearly written and should be modified.

Line 56, should L be described as an RNA-dependent RNA polymerase rather than RNAse?

Lines 104-107, this reads as conjecture and should be modified or justified with appropriate references.

Results, abbreviations should be corrected. 

The authors refer to Negri bodies and endosomes, are these terms being used interchangeably? This should be clarified.

The authors jump between referring to Rabies virus and HEP-Flurry virus in the text and figures. This is confusing and should be amended.

Author Response

Dear,

Thank you for your letter and for the reviewers’ comments concerning our manuscript entitled “Function of host protein Staufen1 in rabies virus replication “(viruses-1199011). Those comments are all valuable and very helpful for revising and improving our paper, as well as the important guiding significance to our research. We have studied comments carefully and have made the correction which we hope meet with approval. The revised portion is marked in red in the paper. Attached please find our response to you.

Reviewer 2 Report

The data was presented clearly.  However the English needs work and the manuscript contains many typos.  For example:

Line 12: should read 'remains a huge...' not 'remains ns a huge...'

Line 18: The sentence beginning ' Downregulating or upregulating...' needs structure.

Line 23: should read '...N protein and G protein...'

Line 45/46: Would make more sense as two sentences. '...genomic RNA to form the ribonucleoprotein complex (RNP). This allows the cellular nuclease P protein to play a crucial...'

Line 100: Delete 'is'. Should read '...(N) protein part...'

Line 107: delete 'it is' and replace with '...assembly, but remains unclear...'

Line 110/111: should read '...N protein and is recruited to the Negri body, an important site...

Line 124: should read '...293T cell total RNA...'

Line 135: Replace with 'Cells were trypsin digested, passaged, and supplemented...'

Section 2.5 Western Blot:  The entire section needs to be re-written to match the other sections. There are many grammatical errors.

Line 203: should read '...aggregated to and co-localized...'

Line 270/271: should read '...composition of the Negri body and especially the on of host factors there during RABV...

Line 318-319: should read '...RNA virus, does not produce viral miRNAs, and instead produces small ncRNAs...'

Line 322:  Is this a question or are you stating that there is a link?

Line 324: Omit 'And'

Line 330/331/332: Remove the duplicated section.

Inconsistencies with terms like UTR, Alu, STAU1 or Staufen1.  Need to go through and correct these inconsistencies.

Author Response

Dear,

Thank you for your letter and for the reviewers’ comments concerning our manuscript entitled “Function of host protein Staufen1 in rabies virus replication “(viruses-1199011). Those comments are all valuable and very helpful for revising and improving our paper, as well as the important guiding significance to our research. We have studied comments carefully and have made the correction which we hope meet with approval. The revised portion is marked in red in the paper. Attached please find our response for your comments. Thank you.

Round 2

Reviewer 1 Report

The authors have improved the introduction and text in line with the detailed comments from the other reviewer and it reads better. However, the authors have not included any further experiments to support their conclusions. 

As highlighted by the authors in their response, testing of over expression of Staufen 1 is a requisite for examining its role as an antiviral protein as concluded. Over expression may not lead to a greater effect on viral replication as demonstrated in the authors own previous work which they cited in there response (Cellular chaperonin CCTγ contributes to rabies virus replication during infection.) but in the current work the authors cannot conclude this as they clearly acknowledge that addition of the large GFP tag disrupts localisation of the over expressed protein. Over expression with a small tag should be performed to test for this possibility. The authors should look in cells that have depleted or over expressed Staufen 1 (detected by IF, as available in the authors lab) and examine RABV replication in those cells, also by IF, similar to the experiments already performed by the authors. The current bulk analysis of cells by WB following siRNA treatment and the relatively heterogenous expression of the GFP-tagged Staufen 1 evident in Figure 2A make drawing conclusions from the current data unreliable. If more detailed analysis is performed, the impact of Staufen 1 may be reliably demonstrated.

The authors posit different reasons why Staufen 1 mRNA may increase during infection in the discussion. They suggest that sequestration of Staufen 1 to the NB may result in the cell compensating to produce more Staufen 1 to maintain cellular function. Is there any evidence that Stafuen 1 has such a regulatory feedback loop- even in the absence of infection, which itself is likely to severely disrupt cellular homeostasis? This should be discussed and citations provided.

On line 199 of the revised text the authors maintain the assertion that colocalisation of Staufen 1 with NB's means that it is important for regulating replication. There is still no data presented in the paper to support this conclusion. The authors reference a paper in which TLR3 was found to localise with NB's and knock down of TLR3 resulted in a similar decrease in RABV titre (Toll-Like Receptor 3 (TLR3) Plays a Major Role in the Formation of Rabies Virus Negri Bodies. PLoS Pathog, 2009). In the referenced paper the authors demonstrate that knock down of TLR3 disrupts NB formation and look at the single cell level by IF to see the impact of the cellular factor on viral replication (similar to what is suggested above), which is not performed in the current manuscript.